# A Novel Germline Frameshift Variant in the Tumor Suppressor Gene *OBSCN* in a Melanoma Patient

**DOI:** 10.3390/ijms262110553

**Published:** 2025-10-30

**Authors:** Barbara Anna Bokor, Aliasgari Abdolreza, Margit Pál, Zita Battyani, Márta Széll, Nikoletta Nagy

**Affiliations:** 1Department of Medical Genetics, University of Szeged, 6720 Szeged, Hungary; 2HUN-REN-SZTE Functional Clinical Genetics Research Group, University of Szeged, 6720 Szeged, Hungary; 3Mór Kaposi Teaching Hospital, 7400 Kaposvár, Hungary

**Keywords:** melanoma, OBSCN, germline variant, cancer predisposition, next-generation sequencing

## Abstract

Malignant melanoma is a complex malignancy with genetic, environmental, and lifestyle factors in its etiology. While germline variants in melanoma predisposition genes have been described, many patients remain genetically unexplained after panel testing. We previously analyzed a Hungarian melanoma cohort (n = 17), identifying variants in predisposing or susceptibility genes in 58.82% of patients. For individuals negative on this melanoma-specific panel, we expanded testing to a 19-gene panel associated with multiple cancer types. Next-generation sequencing was performed, followed by Sanger sequencing for confirmation. Variants were classified according to ACMG guidelines. In a 58-year-old female patient with a history of primary cutaneous melanoma, we identified a novel heterozygous frameshift variant in the tumor suppressor gene *OBSCN* (c.21322_21323insCTGG, p.G7108AfsTer10; NM_001386125.1). This insertion introduces a premature stop codon in exon 89 within the immunoglobulin-like domain, predicting protein truncation. Classified as likely pathogenic (PVS1, PM2), the variant is absent from population databases. To date, somatic *OBSCN* mutations have been reported in melanoma. This first report of a germline *OBSCN* frameshift variant in melanoma expands the genetic landscape of melanoma predisposition and suggests that *OBSCN* may represent a candidate gene contributing to melanoma risk.

## 1. Introduction

Malignant melanoma is a complex disease with a multifactorial etiology, involving genetic, environmental, and lifestyle-related factors that jointly contribute to its development and progression. The incidence of melanoma has been steadily rising worldwide, with approximately 331,722 new cases and 58,667 deaths estimated globally in 2022, according to GLOBOCAN data [1]. The burden is particularly high in populations of European ancestry, and the mortality rate remains substantial due to the tumor’s high metastatic potential. A large body of evidence has accumulated over the last three decades supporting that inherited predisposition contributes to a proportion of melanoma cases, particularly in families with multiple affected members or patients presenting with melanoma at a relatively young age [2].

From the perspective of germline genetic susceptibility, melanoma can arise due to variants in high-penetrance predisposing genes, as well as medium- or low-penetrance susceptibility alleles [3]. Classical examples of high-penetrance melanoma predisposition genes include *CDKN2A* and *CDK4*, both of which regulate critical pathways of cell cycle control and senescence. In addition, genes involved in telomere maintenance (*TERT*, *POT1*, *ACD*, *TERF2IP*) and DNA repair pathways (*BAP1*, *BRCA1*, *BRCA2*, *TP53*) have been implicated in rare but strongly predisposing familial cases. On the other end of the spectrum, common variants of genes such as *MC1R*, *TYR*, *OCA2*, *SLC45A2*, and others contribute modestly to melanoma risk, often in the context of pigmentation phenotype and UV sensitivity [4]. The identification of pathogenic variants in such genes allows for individualized risk assessment, genetic counseling, and tailored surveillance of patients and their family members. Genetic testing panels have become increasingly important in clinical practice, as they allow simultaneous screening of multiple candidate genes associated with melanoma predisposition [5].

In our previous study, we analyzed a Hungarian cohort of 17 melanoma patients with increased risk and a personal or family history suggestive of genetic predisposition. Using a targeted next-generation sequencing panel comprising known melanoma-predisposing and melanoma-susceptibility genes, we identified germline genetic variants in 10 of the 17 patients (58.82%) [6]. These findings underscore the clinical relevance of panel testing in uncovering actionable germline variants in a significant proportion of high-risk melanoma patients. However, in the subset of patients who tested negative for pathogenic or likely pathogenic variants in this established melanoma panel, the question remained whether genes not traditionally associated with melanoma, but linked to other cancer types, might harbor relevant variants. This prompted us to broaden our analysis.

For those patients who were negative on the melanoma-specific panel, we expanded our investigation to include a broader spectrum of cancer-associated genes. To explore potential germline contributors beyond well-known germline melanoma genes, we analyzed a virtual panel of 19 genes selected based on their reported involvement in multiple tumor types and putative tumor suppressor or DNA repair roles. This strategy allowed the inclusion of genes implicated in other cancers, but not systematically investigated in melanoma. Specifically, we designed a virtual panel of 19 genes previously associated with the development of multiple different cancer types: *ABCA1*, *ADAMTSL3*, *ATP8B1*, *CUBN*, *DIP2C*, *EGFL6*, *EPHA3*, *EPHB6*, *FBXW7*, *FLNB*, *GNAS*, *MACF1*, *MLL3*, *OBSCN*, *PKHD1*, *SPTAN1*, *SYNE1*, *TECTA*, and *ZNF668* [7]. Targeted next-generation sequencing was performed, followed by rigorous bioinformatics analysis. Candidate variants of potential clinical relevance were subsequently validated through bidirectional capillary Sanger sequencing, ensuring high confidence in the detected variants.

The aim of this study was to investigate whether an expanded multi-cancer gene panel could identify novel germline variants potentially contributing to melanoma predisposition in patients negative for established high, medium and low-penetrance melanoma genes, and to report the discovery of a novel *OBSCN* truncating variant in these patients.

## 2. Results

### 2.1. Identification of a Novel OBSCN Variant

Within this expanded gene panel analysis, we identified a novel heterozygous germline variant of the tumor suppressor gene obscurin (*OBSCN*) in a 58-year-old female patient with a history of one primary cutaneous melanoma (Figure 1).

The detected variant is described as c.21322_21323insCTGG, p.G7108AfsTer10 (NM_001386125.1) (Figure 2).

This frameshift insertion occurs in exon 89, introducing a premature stop codon after amino acid position 7118 and leading to a truncated protein (Figure 3).

The patient’s personal cancer history included only melanoma, with no evidence of other tumor types. She reported no known family history of melanoma, though her mother was found to carry the same *OBSCN* variant upon genetic testing. The patient’s father was not available for testing, having passed away previously. At present, no additional cancer diagnoses have been recorded among her first-degree relatives (Figure 1).

### 2.2. Figures, Tables and Schemes

According to the American College of Medical Genetics and Genomics (ACMG) guidelines, the variant was classified as likely pathogenic, based on the following criteria:PVS1 (very strong evidence): A null variant (frameshift/nonsense) in a gene where loss of function is a known disease mechanism.PM2 (moderate evidence): The variant is absent, or extremely rare, in large population databases such as gnomAD, suggesting it is not a common benign polymorphism.

“In silico prediction tools supported the deleterious effect of the frameshift variant. The CADD PHRED-scaled score was 37, indicating a high likelihood of pathogenicity. MutationTaster predicted the variant as ‘disease-causing,’ and PROVEAN classified it as ‘deleterious.’ These computational data further support the classification of the OBSCN c.21322_21323insCTGG, p.G7108AfsTer10 variant as likely pathogenic according to ACMG guidelines.”

No functional studies on this particular variant exist to date; however, its predicted truncating nature strongly supports a deleterious effect on protein function.

## 3. Discussion

The *OBSCN* gene, located at 1q42.13, encodes obscurin, a very large cytoskeletal protein that belongs to the family of giant sarcomeric signaling proteins [8]. Obscurin contains multiple immunoglobulin-like (Ig-like) and fibronectin type III (FnIII) domains, in addition to signaling motifs such as RhoGEF and kinase domains [9]. It plays a fundamental role in cytoskeletal organization, cell adhesion, cell–cell recognition, and intracellular signaling pathways [10]. OBSCN is one of the largest genes in the human genome, and its extensive length inherently increases the probability of acquiring both somatic and germline mutations [11].

The identified frameshift variant (p.G7108AfsTer10) is located within the Ig-like domain spanning residues 7077–7146 (Figure 1). Given the truncation within the immunoglobulin-like domain, it is plausible that the variant may compromise cytoskeletal stability and intracellular signaling organization, consistent with previously observed roles of obscurin in maintaining structural integrity and signal transduction. Although the truncating variant results in the loss of C-terminal signaling and kinase regions, several N-terminal domains, including the pleckstrin homology (PH) and Src homology 3 (SH3) domains, remain intact. The retention of these interaction motifs could allow partial binding to cytoskeletal partners but may result in an aberrant or non-functional protein complex. Alternatively, a dominant-negative effect cannot be excluded, whereby the truncated obscurin competes with the full-length protein for binding sites, potentially perturbing cytoskeletal and adhesion-related pathways.

Increasing evidence indicates that *OBSCN* acts as a tumor suppressor gene across multiple cancer types [12,13,14]. Loss or mutation of *OBSCN* has been implicated in: brain tumors, oral squamous cell carcinoma, gastrointestinal tract cancers, Wilms tumor, renal cell carcinoma, female reproductive cancers (ovarian, endometrial), prostate cancer, breast cancer and melanoma. The mechanism of tumorigenesis is thought to involve impaired cytoskeletal stability, altered cell–cell adhesion, and dysregulation of intracellular signaling, ultimately promoting invasive and metastatic phenotypes. Loss of heterozygosity (LOH) at the *OBSCN* locus has been reported in several tumor contexts, supporting its role as a bona fide tumor suppressor.

To date, the literature on *OBSCN* in melanoma remains limited. Previous studies have reported only a somatic missense mutation (p.E4574K), located in the Fn-III 60 domain of the OBSCN protein, in melanoma tumor tissue [7]. No germline *OBSCN* variants have been previously associated with melanoma predisposition. Our finding therefore represents the first report of a nonsense germline likely pathogenic *OBSCN* variant in a melanoma patient. In the present case, no other relevant germline variant was identified in any of the known high, medium or low-penetrance melanoma genes, suggesting that the identified *OBSCN* variant may represent the only identified germline genetic factor, from which we can suppose that it might contribute to melanoma. The proband’s mother, who carries the same variant but remains cancer-free, may reflect incomplete penetrance or the influence of protective genetic and environmental modifiers. This highlights the need for future segregation and functional studies to clarify *OBSCN*’s contribution to melanoma susceptibility. To note, OBSCN expression is not prognostic in melanoma according to TCGA and Protein Atlas data; rare truncating germline OBSCN variants could still contribute to melanoma susceptibility.

Given the truncating nature of this variant, its impact on protein function is expected to be more severe than the previously reported somatic missense substitution. The germline occurrence raises the possibility that *OBSCN* may play a role not only in tumor progression but also in individual susceptibility to melanoma development.

The identification of a novel germline *OBSCN* variant in a melanoma patient has several potential implications:

Expanded gene panels: This case supports the clinical relevance of using expanded multi-cancer panels in melanoma patients with negative standard results, as it enabled the identification of a novel OBSCN variant that would otherwise remain undetected. 

Genetic counseling: Long-term dermatologic and oncologic follow-up is warranted for both the proband and her mother, considering the germline nature of the variant and its potential association with multi-tumor risk.Surveillance: Given the involvement of OBSCN in breast, gastrointestinal, and gynecologic cancers, carriers might benefit from comprehensive surveillance protocols including annual dermatologic examinations, mammography or breast MRI starting at age 40, and colonoscopic screening per standard population guidelines.Research directions: Future research should focus on both predictive and functional validation. Immediate steps include comprehensive in silico modeling and gene network analyses to identify pathways potentially affected by OBSCN loss. Definitive insights, however, will require in vitro studies assessing cytoskeletal organization and cell adhesion in melanoma cells expressing truncated OBSCN.

## 4. Materials and Methods

Genomic DNA was extracted from venous blood mixed with the anticoagulant EDTA using the DNeasy^®^ Blood & Tissue Kit (QIAGEN, Hilden, Germany), as described in the manufacturer’s instructions. For quantification Qubit Fluorometric Quantification instrument was used according to the manufacturer’s instructions.

In this study, all samples were investigated using whole-exome sequencing (WES). The analysis presented in this manuscript, however, focuses on a 19-gene panel extracted from the WES data. This is a virtual panel and was designed to investigate genes with known relevance across multiple cancer types. Thus, the reported findings highlight variants within this virtual gene panel. Targeted next-generation sequencing (NGS) was performed with a virtual cancer gene panel comprising 19 genes associated with multiple tumor types (*ABCA1*, *ADAMTSL3*, *ATP8B1*, *CUBN*, *DIP2C*, *EGFL6*, *EPHA3*, *EPHB6*, *FBXW7*, *FLNB*, *GNAS, MACF1*, *MLL3*, *OBSCN*, *PKHD1*, *SPTAN1*, *SYNE1*, *TECTA*, and *ZNF668)* [4]. Genotypes of patients were determined using next-generation sequencing. Library preparation was carried out using the SureSelectQXT Reagent Kit (Agilent Technologies, Santa Clara, CA, USA). Pooled libraries were sequenced on an Illumina NextSeq 550 NGS platform using the 300-cycle Mid Output Kit v2.5 (Illumina, Inc., San Diego, CA, USA). Adapter-trimmed and Q30-filtered paired-end reads were aligned to the hg19 Human Reference Genome using the Burrows–Wheeler Aligner (BWA). Duplicates were marked using the Picard software package. The Genome Analysis Toolkit (GATK) was used for variant calling (BaseSpace BWA Enrichment Workflow v2.1.1, with BWA 0.7.7-isis-1.0.0, Picard: 1.79 and GATK v1.6-23-gf0210b3). The mean on-target coverage achieved from sequencing was 71 per base, with an average percentage of targets covered greater or equal to 30 of 96% and 90%, respectively. Variants passed by the GATK filter were used for downstream analysis and annotated using the ANNOVAR software tool (version 2017 July 17). Single-nucleotide polymorphism testing was performed as follows: high-quality sequences were aligned with the human reference genome (GRCh37/hg19) to detect sequence variants, and the detected variations were analyzed and annotated. Variants were filtered according to read depth, allele frequency and prevalence in genomic variant databases, such as ExAc (v.0.3) and Curr. Issues Mol. Biol. 2023, 45 5302 Kaviar. Variant prioritization tools (PolyPhen2, SIFT, LRT, Mutation Taster, and Mutation Assessor) were used to predict the functional impact. For variant filtering and interpretation, VarSome and Franklin bioinformatic platforms [https://franklin.genoox.com, accessed on 18 May 2023] were used according to the guidelines of the American College of Medical Genetics and Genomics (ACMG), and population frequencies were checked in the gnomAD database. Candidate variants were confirmed by bidirectional capillary Sanger sequencing.

## 5. Conclusions

In summary, we identified a novel germline frameshift variant in the OBSCN gene (c.21322_21323insCTGG, p.G7108AfsTer10) in a Hungarian patient with malignant melanoma. The variant introduces a premature stop codon within an immunoglobulin-like domain of obscurin, likely resulting in protein truncation and loss of function. Given the established tumor suppressor role of OBSCN and its involvement in cytoskeletal organization, cell adhesion, and intracellular signaling, this alteration may have functional consequences relevant to melanomagenesis.

While OBSCN mutations have been described in several malignancies, including breast, gastrointestinal, and genitourinary cancers, no germline variants have previously been associated with melanoma predisposition. Our observation therefore broadens the mutational and functional spectrum of OBSCN and raises the possibility that its disruption contributes to melanoma susceptibility through impaired cytoskeletal signaling and cellular homeostasis.

Further studies in larger patient cohorts, as well as functional assays, will be necessary to confirm the pathogenicity and penetrance of OBSCN germline variants in melanoma. Nevertheless, this case underlines the value of using extended multi-cancer gene panels in genetically unexplained melanoma cases. Such approaches enhance the detection of rare but potentially relevant variants, thereby advancing our understanding of melanoma genetics and informing future risk assessment, genetic counseling, and precision oncology efforts.

## Figures and Tables

**Figure 1 ijms-26-10553-f001:**
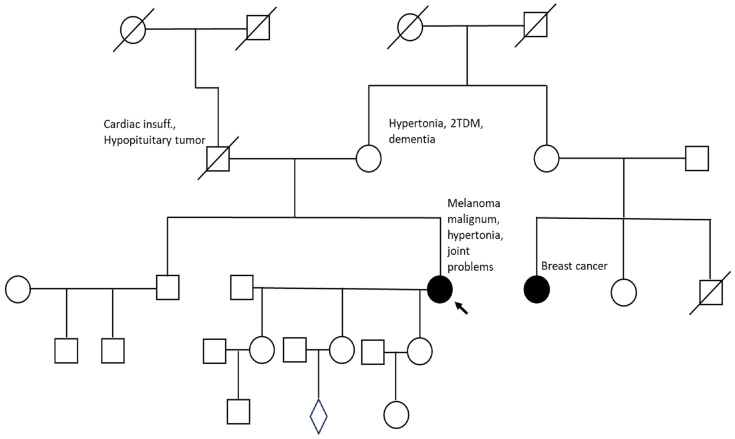
Pedigree of the Hungarian melanoma patient carrying the novel germline *OBSCN* variant. The proband (arrow) was diagnosed with a primary cutaneous melanoma and found to carry the heterozygous frameshift variant c.21322_21323insCTGG (p.G7108AfsTer10). The patient’s mother also carries the same variant but has no cancer history to date. The father was not available for testing (deceased). Squares and circles represent males and females, respectively. Filled symbols denote individuals affected by cancer. Unfilled symbols denote individuals with no known cancer diagnosis. Diagonal lines indicate deceased individuals. The arrow marks the proband, the individual in whom the genetic evaluation was initiated. A horizontal line connecting a square and a circle represents a biological parental relationship, and vertical lines show offspring.

**Figure 2 ijms-26-10553-f002:**
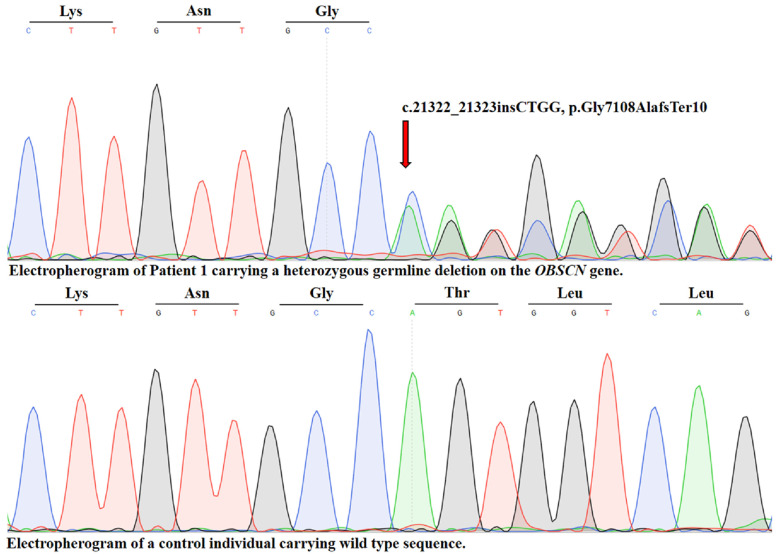
Sanger sequencing confirmation of the novel *OBSCN* variant. Bidirectional capillary sequencing verified the heterozygous frameshift insertion (c.21322_21323insCTGG, p.G7108AfsTer10) in exon 89 of the *OBSCN* gene (NM_001386125.1). The electropherogram demonstrates the site of insertion (arrow), confirming the presence of the variant in the proband. Colored peaks: represent the four nucleotides detected during capillary electrophoresis: Adenine (A) = green, Cytosine (C) = blue, Guanine (G) = black and Thymine (T) = red.

**Figure 3 ijms-26-10553-f003:**
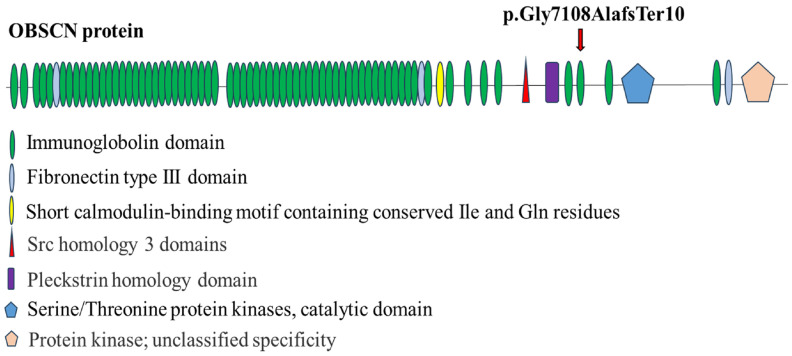
Schematic representation of the OBSCN protein domains and the location of the novel variant. The OBSCN protein comprises multiple structural and signaling domains, including immunoglobulin-like repeats, fibronectin type III domains, RhoGEF, and kinase domains. The novel c.21322_21323insCTGG, p.G7108AfsTer10 germline frameshift variant (arrow) identified in the proband is indicated in exon 89, within the immunoglobulin-like domain (amino acids 7077–7146). The resulting premature stop codon is predicted to truncate the protein, potentially disrupting cytoskeletal organization and cellular signaling (SMART https://smart.embl.de/smart/show_motifs.pl, accessed on 20 March 2025).

## Data Availability

The data presented in this study are available from the corresponding author upon request. The data are not publicly available because they are genetic data.

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
