# Peer review of "A Novel Germline Frameshift Variant in the Tumor Suppressor Gene OBSCN in a Melanoma Patient"

_ijms, 2025, doi:10.3390/ijms262110553_

Round 1
Reviewer 1 Report
Comments and Suggestions for Authors
In this study, Bokor et al. identifies a novel germline frameshift variant in the OBSCN gene in a patient with a history of melanoma. The authors used targeted sequencing using predefined gene panel and validation by Sanger sequencing. I have listed my comments on this manuscript below:
- I am troubled by the size of the panel used to identify novel germline mutation/s. The best unbiased approach would be to run at the very least, exome sequencing.
- OBSCN is a very big gene. This automatically increases the possibility of acquiring mutations. According to Mendiratta et al. (Nature communications 2021) in study titled “Cancer gene mutation frequencies for the U.S. population”, OBSCN (9%) is the second most mutated kinase behind TTN (30%). This is by their sheer size. COSMIC Cancer Gene Census list does not include these genes as drivers. Any acquisition of somatic or germline mutation have a tendency to put a selective pressure. This is a very important point brought up in the field many times before.
- The sample size is very limited. Only 1 out of 17 patients harbored a deleterious mutation in this gene, which could easily occur by chance given the ~9% probability mentioned above. Additionally, the authors fail to provide any details on the pathology or genomic landscape of the melanoma from this patient. Just having a history is not enough to link it to this mutation. OBSCN is more relevant in other cancer types. Do the authors see anything in the mutational profile of the melanoma from this patient that they deemed remarkable enough to link it to this mutation? Demonstrating the role of this gene in tumorigenesis in this patient is therefore essential.
- A much wider panel of targeted cancer relevant genes should be used if that is the route the authors prefer. Many medical institutes have such panels that is not focused only one cancer type and identify both somatic and germline variants.
- The vast TCGA data does not show a strong pathogenic role of this gene in melanoma. Kaplan-Meier survival plots for OBSCN expression in melanoma show no substantial difference in survival between patients with high and low OBSCN expression. They explicitly mention that "OBSCN is not prognostic in skin cutaneous melanoma" (https://www.proteinatlas.org/ENSG00000154358-OBSCN/cancer/melanoma)
Author Response
"Please see the attachment."

Reviewer 2 Report
Comments and Suggestions for Authors
In this case report the authors described the first identification of a novel germline nonsense variant in the OBSCN gene in a patient with melanoma, suggesting a potential role for this gene in inherited melanoma predisposition and highlighting the value of expanded multi-cancer gene panels.
There are critical issues related to the clarity of the clinical data, methodological justification, and scientific support of the claims must be addressed to enhance the manuscript's impact and scientific rigor.
Abstract and Introduction
- Vague Abstract Statement (Lines 27–28): The phrase "This first report of a germline OBSCN nonsense variant in melanoma suggests that OBSCN may contribute to melanoma predisposition..." is vague.
- Melanoma Contextualization (Lines 37, 41, 50, 55): The introduction must better contextualize the genetic and epidemiological background of melanoma. Provide the missing reference for Line 37 (epidemiological data, e.g., GLOBOCAN data). The newer reference (DOI: 10.3390/cancers17111784) should be used to improve the contextualization of melanoma genetics (Line 50). The misplaced Reference 1 can be repositioned at the end of Line 41. Reference is missing for Line 55.
- Missing Aim Statement and Result: The Introduction lacks a clear Statement of Purpose (Aim Statement) and does not sufficiently highlight the salient finding.
- Arbitrary Gene Selection Rationale: The selection of the 19-gene expansion panel appears arbitrary without a clear biological rationale.
Results and Functional Support
- Insufficient Functional Justification (Lines 120–121): The statement that the variant's truncating nature supports a deleterious effect is insufficient, even for a Case Report. The authors must incorporate predictive bioinformatic tool data (e.g., CADD, PolyPhen-2, SIFT). They should state the specific tool(s) used and present the resulting pathogenicity scores to scientifically support the classification as likely pathogenic.
- Figure 3 Generation: The method for generating the schematic protein domain map is missing.
- Improper Terminology (Figure 2 Caption): The use of the term "Sequenogram" in the Figure 2 caption is inappropriate.
Discussion and Clinical Coherence
- Missing References in Discussion (Lines 124–129, 142): Multiple statements regarding the structure, function, and role of OBSCN as a TSG lack references.
- Hypothetical Speculation (Lines 131–133): The discussion about the variant affecting cellular adhesion and "signaling fidelity" without in vitro or in vivo validation is hypothetical. The phrase "signaling fidelity" must be clarified or replaced.
- Incomplete Truncation Analysis: The authors must also discuss the functional consequences of the retained domains. Discuss the possible impact of the preserved PH and SH3 domains on protein function (e.g., does it suggest a non-functional truncated protein or a potential dominant-negative effect?), providing a more nuanced analysis of the truncating variant.
- Unsubstantiated Causal Role (Lines 146–152): The strong claim that OBSCN plays a role in melanoma pathogenesis (Lines 149–152) must be supported, or qualified.
- Action Required: The authors must disclose any other predisposition/developmental mutations identified in the proband (e.g., in other melanoma-related genes like CDKN2A, etc.) to assess the context of the OBSCN variant. Furthermore, the difference in phenotype (melanoma in proband vs. apparently healthy mother) must be discussed in terms of incomplete penetrance or the role of modifying genes.
- Lack of Coherence in Clinical Points. The four points (Lines 155-166) lack supporting details from the case itself.
- Point 1 (Expanded Panels): This is a general principle. The Case Report should use its own finding to justify the need for such panels.
- Point 2 (Follow-up): The discussion of longitudinal follow-up for the proband is sensible due to her diagnosis
- Point 3 (Surveillance): The authors must provide specific, actionable surveillance suggestions for a carrier, based on the documented multi-tumor risk (e.g., specific screening for breast and gastrointestinal cancer) they depicted, especially given the sister's breast cancer diagnosis (which must be integrated into the discussion).
- Point 4 (Research Directions): Since the authors reported a truncating variant in a candidate tumor suppressor gene (OBSCN) with strong clinical suspicion but no direct functional proof, the necessary experiments fall into two main categories: Bioinformatic/Predictive (immediate, in silico proof) and Molecular/Cellular (definitive, in vitro proof).
Materials and Methods
- Missing Protocol Details: The description of DNA extraction is too vague. Specify the standard protocol or kit used for Genomic DNA extraction from peripheral blood leukocytes.
- Missing Bioinformatics Tools: The description of the analysis is incomplete.
- Action Required: Provide the specific names of the bioinformatics tools/pipelines used for variant calling, annotation, and interpretation.
Conclusion
The Conclusions section, which summarizes the findings and their implications, lacks the necessary scientific support.
Author Response
"Please see the attachment."

Round 2
Reviewer 2 Report
Comments and Suggestions for Authors
N/A
Author Response
Thank you for reviewing the manuscript. I did not find any additional comments or suggestions to answer.